# TAIPAN: EFFICIENT AND EXPRESSIVE STATE SPACE LANGUAGE MODELS WITH SELECTIVE ATTENTION

## ABSTRACT

Efficient long-context language modeling remains a significant challenge in Natural Language Processing (NLP). While Transformers dominate language tasks, they struggle with long sequences due to quadratic computational complexity in training and linearly scaling memory costs during inference. Recent State Space Models (SSMs) such as Mamba offer alternatives with constant memory usage, but they underperform in tasks requiring extensive in-context retrieval. We introduce Taipan, a novel hybrid architecture that combines Mamba-2 with Selective Attention Layers (SALs). These SALs identify tokens requiring long-range interactions, remove less important features, and then augment their representations using the attention module. This approach balances Mamba's efficiency with Transformer-like performance in memory-intensive tasks. By constraining the attention budget, Taipan extends accurate predictions to context lengths of up to 1 million tokens while preserving computational efficiency. Our experiments demonstrate Taipan's superior performance across various scales and tasks, offering a promising solution for efficient long-context language modeling.

## 1 INTRODUCTION

Transformer-based architectures Vaswani (2017); Brown (2020) have revolutionized Natural Language Processing (NLP), delivering exceptional performance across diverse language modeling tasks Touvron et al. (2023). This success stems from their ability to capture complex word dependencies using the self-attention mechanism. In addition, Transformers are highly scalable and well-suited for parallel training on large datasets. However, despite their success, they still face notable challenges when handling long-context sequences. Specifically, the self-attention mechanism suffers from quadratic computational complexity, and the memory requirement grows linearly with context length during inference, as the model must store key-value vectors for the entire context. These factors impose practical constraints on sequence length due to the high computational and memory costs.

To this end, recent advancements in recurrent-based architectures, particularly State Space Models (SSMs) Gu et al. (2021b;a), have emerged as promising alternatives for efficient language modeling Gu & Dao (2023); Dao & Gu (2024). SSMs offer constant memory usage during inference, and architectures like Mamba-2 Dao & Gu (2024), a variant of SSMs, have demonstrated performance comparable to Transformers in certain language tasks Waleffe et al. (2024). Some studies even suggest that SSMs can outperform Transformers in areas like state tracking Merrill et al. (2024) due to their Markovian nature. However, despite these advancements, SSM-based models still fall short in scenarios requiring in-context retrieval or handling complex long-range dependencies Arora et al. (2024); Waleffe et al. (2024).

To address these challenges, we introduce Taipan, a hybrid architecture that combines the efficiency of Mamba with enhanced long-range dependency handling through Selective Attention Layers (SALs). While Mamba is highly efficient, it relies on the Markov assumption—where predictions are based solely on the last hidden state—which can lead to information loss for tokens that need interactions with distant tokens. To mitigate this, Taipan incorporates SALs that strategically select key tokens in the input sequence requiring long-range dependencies. These selected tokens first undergo feature refinement to remove unimportant information, and then are passed through an attention module to capture long-range dependencies. Less critical tokens bypass the attention step, as

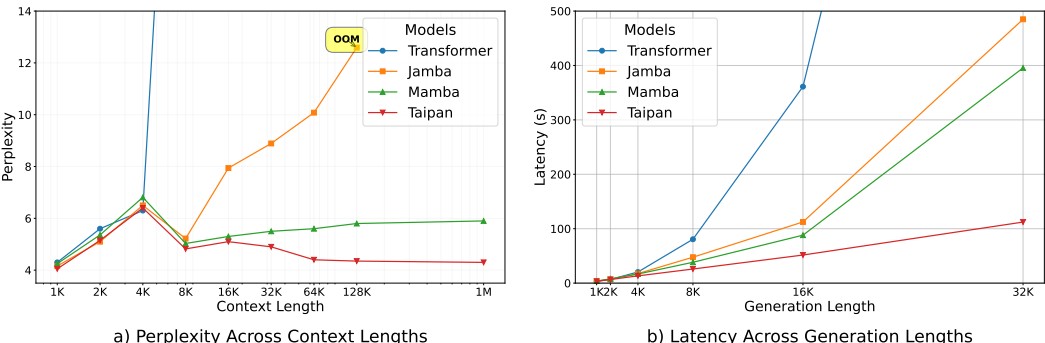

Figure 1: **Model Performance Comparison**. a) Perplexity across different context lengths. Lower perplexity indicates better performance. b) Latency comparison of models at various generation lengths. Taipan exhibits significantly lower latency and superior scaling compared to other strong baselines for longer sequences.

we hypothesize that their Markovian representations from Mamba contain sufficient information for accurate prediction, obviating the need for additional attention-based augmentation. This selective approach enables Taipan to balance Mamba's computational efficiency with enhanced long-range modeling capabilities.

SALs play a crucial role in Taipan's design, both in enhancing performance and ensuring computational efficiency. By focusing the attention mechanism on a subset of important tokens, SALs reduce the computational costs that come from attention modules. This targeted approach enables Taipan to excel in memory-intensive tasks while maintaining efficiency during both training and inference. Importantly, Taipan retains the linear memory usage characteristic of SSMs, offering a significant advantage over traditional Transformer models in handling extremely long sequences.

We scale Taipan to 190M, 450M, and 1.3B parameters, pre-training on 100B tokens. Experimental results demonstrate Taipan's superior performance across a wide range of tasks. In zero-shot language modeling evaluations, Taipan consistently outperforms both Transformer and Mamba baselines, showcasing its strong general language understanding capabilities. More notably, in memory-intensive tasks such as long-context retrieval and structured information extraction, Taipan exhibits significant improvements over Mamba-2, addressing a key limitation of existing recurrent-based models. Furthermore, Taipan demonstrates remarkable extrapolation capabilities, maintaining high performance on sequences up to 1 million tokens in context-length - while preserving efficient generation capabilities. This combination of broad task proficiency, superior performance in memory-intensive scenarios, and exceptional long-context modeling positions Taipan as a versatile and powerful architecture for advanced language processing tasks.

## 2 BACKGROUND

This section briefly overviews the foundational architectures relevant to our work. We first review Causal Self-Attention Vaswani (2017), the core mechanism of Transformer models. We then discuss Linear Attention Katharopoulos et al. (2020), an efficient variant that achieves linear complexity. Finally, we examine Mamba-2, a recent architecture that generalizes Linear Attention using structured state-space models (SSMs) Dao & Gu (2024). We emphasize how each model balances computational efficiency and recall accuracy, particularly in memory-intensive tasks .

### 2.1 CAUSAL SELF-ATTENTION

Causal Self-Attention is the key component in Transformer architectures that allows each token in a sequence to attend to all other previous tokens (Vaswani, 2017). Given an input sequence $\mathbf{X} = [\mathbf{x}_1, \ldots, \mathbf{x}_L] \in \mathbb{R}^{L \times d}$, where $L$ is the sequence length and $d$ is the embedding dimension,

self-attention firsts computes the query, key, and value vectors for each token via linear projections:

$$\mathbf{q}_i = \mathbf{W}_Q \mathbf{x}_i, \quad \mathbf{k}_i = \mathbf{W}_K \mathbf{x}_i, \quad \mathbf{v}_i = \mathbf{W}_V \mathbf{x}_i$$

where $\mathbf{W}_Q, \mathbf{W}_K, \mathbf{W}_V \in \mathbb{R}^{d \times d}$ are learnable weight matrices.

Then, the attention output $\mathbf{o}_i$ for each token $\mathbf{x}_i$ will be calculated as a weighted sum of the value vectors over the distribution of similarity matrix between its query vector and previous key vectors:

$$\mathbf{o}_i = \sum_{t=1}^{i} \frac{\exp(\mathbf{q}_i^\top \mathbf{k}_t / \sqrt{d})}{\sum_{j=1}^{i} \exp(\mathbf{q}_i^\top \mathbf{k}_j / \sqrt{d})} \mathbf{v}_t$$

The non-linear softmax distribution allows the models to capture intricate relationships between tokens, and concentrate on salient features Qin et al. (2022); Zhao et al. (2019). As such, self-attention can encode complex language patterns and long-range dependencies that are crucial for complex language understanding and generation tasks.

## 2.2 LINEAR ATTENTION

To address the quadratic complexity, recent work has shown that it is possible to achieve linear complexity with the attention mechanism by replacing the softmax attention with dot-product attention (Shen et al., 2021; Katharopoulos et al., 2020). Given a feature transformation $\phi(\mathbf{x})$, causal self-attention can be rewritten as:

$$\mathbf{o}_i = \sum_{t=1}^{i} \frac{\phi(\mathbf{q}_i)^\top \phi(\mathbf{k}_t)}{\sum_{j=1}^{i} \phi(\mathbf{q}_i)^\top \phi(\mathbf{k}_j)} \mathbf{v}_t$$

Then, using the associate property of matrix multiplication, this can be reformulated as:

$$\mathbf{o}_i = \frac{\phi(\mathbf{q}_i)^\top \sum_{t=1}^{i} \phi(\mathbf{k}_t) \mathbf{v}_t^\top}{\phi(\mathbf{q}_i)^\top \sum_{t=1}^{i} \phi(\mathbf{k}_t)}$$

Let $\mathbf{S}_i = \sum_{t=1}^{i} \phi(\mathbf{k}_t) \mathbf{v}_t^\top$ and $\mathbf{z}_i = \sum_{t=1}^{i} \phi(\mathbf{k}_t)$. We can then rewrite the equation in a recurrent form:

$$\mathbf{S}_i = \mathbf{S}_{i-1} + \phi(\mathbf{k}_i) \mathbf{v}_i^\top$$

$$\mathbf{o}_i = \frac{\mathbf{S}_i \phi(\mathbf{q}_i)}{\mathbf{z}_i^\top \phi(\mathbf{q}_i)} \approx \mathbf{S}_i \phi(\mathbf{q}_i)$$

This formulation allows for efficient training and inference. Let $\mathbf{Q}, \mathbf{K}, \mathbf{V} \in \mathbb{R}^{L \times d}$ be the query, key, and value matrices of the sequence input $\mathbf{X}$. During training, we can use the matrix multiplication form: $\mathbf{O} = (\mathbf{Q}\mathbf{K}^\top \odot \mathbf{M}_L)\mathbf{V}$, where $\mathbf{M}_L$ is a causal mask. At inference time, we can use the recurrent form for efficient sequential processing.

However, despite its computational efficiency, linear attention has notable limitations compared to softmax attention. The dot-product approximation in linear attention lacks the nonlinear normalization of softmax, often resulting in a more uniform distribution of attention weights Han et al. (2023). This uniformity can impair the model's ability to focus sharply on specific and relevant tokens. Consequently, linear attention models may underperform in tasks requiring precise in-context retrieval or focused attention on particular input segments Han et al. (2023).

## 2.3 MAMBA-2

Mamba Gu & Dao (2023) is a variant of structured state space models (SSMs) that uses the selective data-dependent mechanism. Mamba-2 Dao & Gu (2024) builds on this foundation, revealing deep connections between SSMs and linear attention Katharopoulos et al. (2020) through the framework of structured state-space duality (SSD).

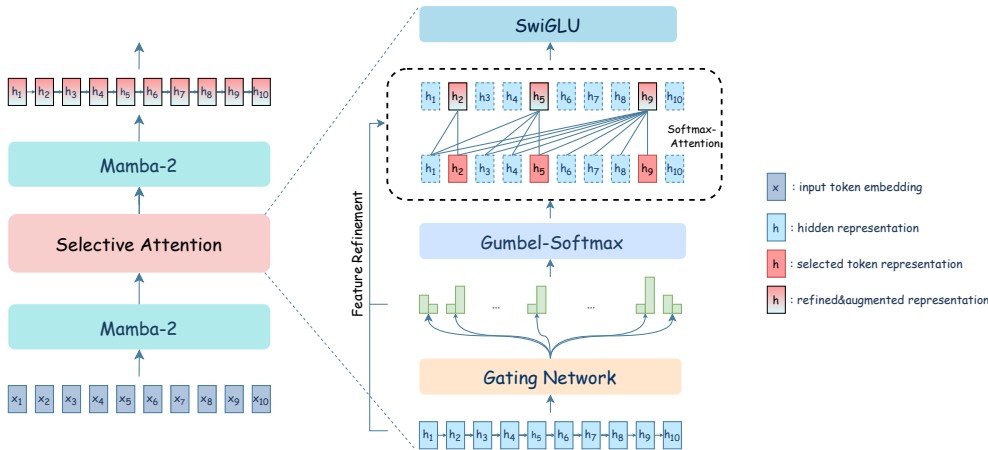

Figure 2: An overview of the Taipan architecture.

The core of Mamba-2 can be defined by using the recurrent form:

$$\mathbf{h}_t = \mathbf{A}_t \mathbf{h}_{t-1} + \mathbf{B}_t \mathbf{x}_t$$
$$\mathbf{o}_t = \mathbf{C}_t \mathbf{h}_t$$

where $\mathbf{A}_t$ is further simplified to a scalar multiplied by the identity matrix. This formulation allows Mamba-2 to be interpreted as a generalization of linear attention.

The key insight of Mamba-2 is that this recurrence can be equivalently expressed as a matrix multiplication:

$$\mathbf{O}_t = (\mathbf{L}_t \odot \mathbf{C}_t \mathbf{B}_t^\top) \mathbf{X}_t$$

where $\mathbf{L}$ is a 1-semiseparable matrix. This matrix form reveals the duality between the recurrent (linear-time) and attention-like (quadratic-time) computations. Also, the 1-semiseparable matrix $\mathbf{L}$ encodes the temporal dependencies, while $\mathbf{CB}^\top$ represents content-based interactions similar to attention. This formulation generalizes linear attention, which can be seen as a special case where $\mathbf{L}$ is the all-ones lower triangular matrix.

While Mamba-2 is efficient, it shares the same limitations as Linear Attention in terms of precise memory recall Arora et al. (2024); Wen et al. (2024), leading to reduced performance in tasks that demand accurate retrieval of specific sections in the input sequence.

## 3 TAIPAN MODEL

To address the limited modeling capabilities of Mamba-2 and Linear Attention while preserving their computational efficiency, we introduce Taipan, a new architecture for sequence encoding in language modeling. In Taipan, we strategically incorporate Selective Attention Layers (SALs) within the Mamba framework, as shown in Figure 2. SALs are inserted after every $K$ Mamba-2 blocks, creating a hybrid structure that combines Mamba-2's efficiency with Transformer-style attention for effective sequence representation.

The core of SALs is a gating network that identifies important tokens for enhanced representation modeling. These tokens undergo two phases: (1) feature refinement to filter out irrelevant information and (2) representation augmentation via softmax attention. This allows Taipan to capture complex, non-Markovian dependencies when necessary.

Taipan processes input through Mamba-2 blocks, with SALs periodically refining key token representations. These enhanced representations are then passed into the subsequent Mamba-2 layers, influencing further processing. This hybrid structure balances Mamba-2's efficiency with the expressive power of SALs, enabling Taipan to excel in tasks requiring both speed and accurate information retrieval. The following sections detail each component's structure and function.

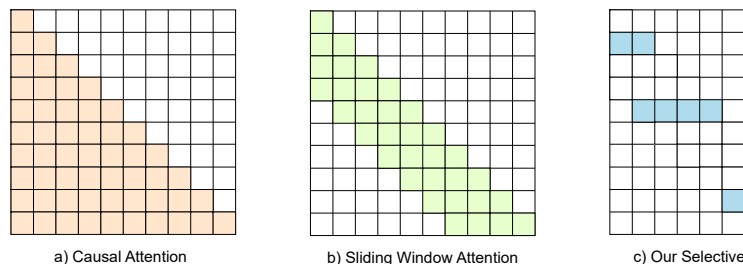

Figure 3: Attention mechanisms in Taipan's Selective Attention Layers. White areas indicate no attention. (a) Full Causal Attention (b) Sliding Window Attention ($w = 4$) (c) Selective Attention ($C = 0.3$, $w = 5$)

## 3.1 SELECTIVE ATTENTION LAYERS

Selective Attention Layers (SALs) are the key innovation in Taipan, designed to enhance the model's ability to focus on critical tokens while maintaining overall efficiency. These layers employ a lightweight gating network $G_\theta$ to dynamically determine which tokens should undergo softmax attention processing.

For each token hidden representation $\mathbf{h}_i$ in the input sequence, the gating network $G$ computes a score vector:

$$\mathbf{s}_i = G_\theta(\mathbf{h}_i) \tag{1}$$

where $G_\theta : \mathbb{R}^d \to \mathbb{R}^2$ is parameterized by $\theta$. This score vector $\mathbf{s}_i = [s_{i,0}, s_{i,1}]$ serves two purposes: 1) it is used to generate a binary mask $m_i$ for token selection, and 2) it guides feature refinement.

To maintain differentiability while allowing for discrete token selection, we employ the Straight-Through Gumbel-Softmax trick Jang et al. (2017). A binary mask $m_i$ is generated from $\mathbf{s}_i$ to select tokens during the forward pass of the network:

$$m_i = \mathrm{argmax}(\mathrm{GumbelSoftmax}(\mathbf{s}_i, \tau)) \tag{2}$$

where $\tau$ is the temperature parameter. $\mathbf{h}_i$ will only be selected for attention processing if $m_i = 1$.

For the backward pass, we instead use continuous Gumbel-Softmax approximation of $m_i$ to achieve computation differentiability for the network:

$$\tilde{m}_i = \frac{\mathbb{I}[m_i = 0] \exp((\mathbf{s}_{i,0} + g_0)/\tau) + \mathbb{I}[m_i = 1] \exp((\mathbf{s}_{i,1} + g_1)/\tau)}{\exp((\mathbf{s}_{i,0} + g_0)/\tau) + \exp((\mathbf{s}_{i,1} + g_1)/\tau)} \tag{3}$$

where $\mathbb{I}[]$ is the indicator function, and $g_0$ and $g_1$ are i.i.d samples from the Gumbel$(0, 1)$ distribution. In this way, we are able to train our entire model, including the gating network, in an end-to-end fashion for language modeling.

For the selected tokens (those with a mask value $m_i$ of 1), we compute their attention-based representations:

$$\mathbf{o}_i = \mathrm{Attention}(\mathbf{q}_i, \mathbf{K}, \mathbf{V}) \tag{4}$$

where $\mathbf{q}_i$ is the query vector for the $i$-th selected token (denoted $\mathbf{h}_i^s$), and $\mathbf{K}$ and $\mathbf{V}$ are the key and value matrices for previous tokens.

In our model, the score vector $\mathbf{s}_i$ is also used to refine the representations of selected tokens. We employ the softmax of $\mathbf{s}_i$ to compute the mixing weights: $[1 - \alpha_i, \alpha_i] = \mathrm{softmax}(\mathbf{s}_i)$. The final output for a selected token $\mathbf{h}_i^s$ is a weighted combination:

$$\mathbf{h}_i^s = (1 - \alpha_i)\mathbf{h}_i^s + \alpha_i \mathbf{o}_i \tag{5}$$

As such, Taipan can adaptively preserve key information in $\mathbf{h}_i^s$ while enriching the representation with the attention output $\mathbf{o}_i$. In other words, $\alpha_i$ acts as the *data-dependent factor*, filtering out unimportant features from the original representation while integrating richer information from the attention outputs. Here, it is important to note that unselected tokens (i.e., $m_i = 0$) skip the attention module and retain their original representations from Mamba-2.

Finally, all token representations are passed through a residual SwiGLU Shazeer (2020) layer:

$$\mathbf{h} = \mathbf{h} + \text{SwiGLU}(\mathbf{h}) \tag{6}$$

This final transformation ensures that all token representations undergo consistent non-linear processing before being passed to the next layer in the network, enhancing the model's ability to capture complex dependencies.

## 3.2 SLIDING WINDOW ATTENTION

To maintain linear time complexity while leveraging the benefits of attention, Taipan employs Sliding Window Attention (SWA) Beltagy et al. (2020). SWA's computational complexity scales linearly with sequence length, allowing Taipan to handle theoretically unlimited context lengths during inference. Importantly, the combination of Selective Attention and Sliding Window Attention in Taipan leads to a significantly sparser attention weight map compared to full attention or standard windowed attention (Figure 3), thus enhancing the computational efficiency of Selective Attention for processing long sequences for our model. In addition, the sparser attention map allows us to afford a longer sliding window (i.e., $w = 2048$ in our work) to effectively capture longer-range dependencies for input sequences. In this way, our designed Taipan architecture offers a mechanism to balance the efficient processing of long sequences with the ability to capture important long-range dependencies, thereby addressing a key limitation of existing efficient attention mechanisms. Finally, removing positional embeddings from the Attention Module improves extrapolation capabilities, suggesting that the model can better generalize temporal relationships. We explore this impact of positional embeddings in more detail in Section 5.2.

## 3.3 TRAINING AND INFERENCE

To better balance efficiency and expressiveness, we introduce an attention budget constraint. Given a predefined budget $C$, representing the desired fraction of tokens to receive attention, we incorporate a constraint loss into our training objective:

$$\mathcal{L}_{\text{constraint}} = \sum_{n=1}^{N} \left\| C - \frac{\sum_{i=1}^{L} m_i}{L} \right\|_2^2 \tag{7}$$

Here, $N$ is the number of SALs, $L$ is the sequence length, and $\sum_{i=1}^{L} m_i$ represents the number of tokens selected for attention processing. During training, we employ the Straight Through Gumbel Softmax estimator for $\tilde{m}_i$ in the backward pass Jang et al. (2017); Bengio et al. (2013), ensuring differentiability while maintaining discrete token selection in the forward pass, thereby enabling end-to-end training of the entire model. As such, our overall training objective includes a standard cross-entropy loss $\mathcal{L}_{\text{CE}}$ for language modeling and the budget constraint term: $\mathcal{L} = \mathcal{L}_{\text{CE}} + \lambda \mathcal{L}_{\text{constraint}}$, where $\lambda$ is a hyperparameter.

During inference, Taipan processes input tokens sequentially through Mamba-2 blocks. At each Selective Attention Layer, the gating network $G_\theta$ computes a score vector $\mathbf{s}_i = G_\theta(\mathbf{h}_i)$ for each token representation $\mathbf{h}_i$. This score computes a binary mask $m_i$ to determine if $\mathbf{h}_i$ should be used for attention processing. Consequently, our selective attention approach maintains Mamba-2's efficiency for most tokens while applying targeted attention to critical elements, enabling effective long-range dependency modeling with minimal computational overhead.

## 4 EXPERIMENTS

We conducted extensive experiments to evaluate Taipan's performance across various scales and tasks. Our evaluation strategy focuses on three main areas: (1) zero-shot evaluation on diverse benchmarks to demonstrate Taipan's general language understanding capabilities (Section 4.2), (2) in-context retrieval tasks to assess Taipan's ability to retrieve information from historical contexts (Section 4.3), and (3) extrapolation ability in long-context scenarios to evaluate performance on extremely long sequences (Section 4.4).

## 4.1 Experimental Setup

We evaluate Taipan across three model sizes: 190M, 450M, and 1.3B parameters. To ensure a comprehensive and fair comparison, we benchmark Taipan against three strong baselines:

- **Transformer++** Touvron et al. (2023): An enhanced version of the LLaMA architecture Touvron et al. (2023), incorporating Rotary Positional Embeddings Su et al. (2024), SwiGLU Shazeer (2020), and RMSNorm Zhang & Sennrich (2019).

- **Mamba-2** Dao & Gu (2024): A state-of-the-art linear RNN model based on State Space Models (SSMs). Each Mamba-2 block consists of a depthwise convolutional layer Poli et al. (2023); Gu & Dao (2023), an SSM layer Dao & Gu (2024), and MLP layers.

- **Jamba** Lieber et al. (2024): A hybrid model combining full Causal Self-Attention layers (with Rotary Position Embedding Su et al. (2024)) and Mamba-2 layers. Unlike Taipan, Jamba uses full Causal self-attention instead of selective attention, retains positional embeddings, and lacks a feature refinement mechanism.

**Implementation Details** We train all models from scratch in three configurations: 190M, 450M, and 1.3B parameters. The training process is consistent across configurations with the following hyperparameters: a batch size of $0.5M$ tokens per step, a cosine learning rate schedule with 2000 warm-up steps, and AdamW Loshchilov (2017) optimization with a peak learning rate of $5e-4$ decaying to a final rate of $1e-5$. We apply a weight decay of $0.01$ and use gradient clipping with a maximum value of $1.0$. All models are trained with a fixed context length of 4096 tokens.

The training data size varies by model scale: the 190M model is trained on 27 billion tokens, while the 450M and 1.3B models are trained on 100 billion tokens. The dataset details can be found in Appendix A.

For Taipan-specific implementation, we use a hybrid ratio of $6:1$, inserting a Selective Attention Layer (SAL) after every 6 Mamba-2 Blocks. The Mamba-2 blocks are kept identical to the original work Dao & Gu (2024). We set the attention capacity $C = 0.15$. The sliding window attention mechanism uses a window size ($w$) of 2048 tokens.

| Params & Data | Model | Wino. | PIQA | Hella. | ARC$_E$ | ARC$_C$ | OB. | Truth. | RACE | BoolQ | Avg. |
|---|---|---|---|---|---|---|---|---|---|---|---|
| 190M 27B | Transformer++ | 47.1 | 60.9 | 27.9 | 42.2 | 20.5 | 18.9 | 42.9 | 25.4 | 57.2 | 38.1 |
| | Mamba | 49.6 | 60.7 | 29.3 | 45.3 | **21.8** | 20.6 | 40.8 | 27.2 | **59.3** | 39.4 |
| | Jamba | 49.9 | 60.3 | 29.2 | 46.3 | 21.4 | 18.5 | 39.8 | **27.4** | 58.6 | 39.1 |
| | **Taipan** | **51.0** | **62.6** | **29.4** | **46.7** | 20.7 | **21.8** | 41.1 | 26.6 | 58.7 | **39.9** |
| 450M 100B | Transformer++ | 51.5 | 67.6 | 42.3 | 60.8 | 27.7 | 33.4 | **39.2** | 30.5 | 54.7 | 45.3 |
| | Mamba | 52.7 | 68.9 | 42.7 | 61.4 | 27.1 | 34.0 | 38.5 | 29.3 | 53.2 | 45.3 |
| | Jamba | **53.1** | 69.3 | 44.3 | 62.6 | 28.7 | 34.4 | 37.5 | 31.3 | 55.7 | 46.3 |
| | **Taipan** | 53.0 | **69.6** | **46.6** | **65.6** | **32.9** | **36.6** | 38.6 | 30.7 | 60.4 | **48.2** |
| 1.3B 100B | Transformer++ | 53.8 | 71.6 | 53.8 | 63.2 | 36.3 | 36.4 | **44.0** | 31.2 | 59.4 | 49.9 |
| | Mamba | 55.2 | 73.0 | 55.6 | 70.7 | 38.0 | 39.0 | 39.9 | 32.0 | **61.8** | 51.7 |
| | Jamba | 54.7 | 73.8 | 55.8 | 69.7 | 37.6 | 41.8 | 40.4 | 32.8 | 59.2 | 51.8 |
| | **Taipan** | **57.0** | **74.9** | **57.9** | **71.2** | **39.3** | 40.4 | 43.0 | **34.4** | 61.5 | **53.3** |

Table 1: Zero shot results of Taipan against baseline models.

## 4.2 Language Modeling Performance

We report the zero-shot performance of Taipan and baseline models on a diverse set of common-sense reasoning and question-answering tasks. These include Winograd (Wino.) Sakaguchi et al. (2021), PIQA Bisk et al. (2020), HellaSwag (Hella.) Zellers et al. (2019), ARC-easy and ARC-challenge (ARCe & ARCc) Clark et al. (2018), OpenbookQA (OB.) Mihaylov et al. (2018), TruthfulQA (Truth.) Lin et al. (2021), RACE Lai et al. (2017), and BoolQ Clark et al. (2019). It is worth noting that these tasks are brief and do not involve in-context learning, thus inadequately demonstrating long-context modeling or in-context learning retrieval abilities.

Table 1 presents the zero-shot results for models of three sizes: 190M, 450M, and 1.3B parameters. The results are evaluated using the lm-evaluation-harness[1] Gao et al. (2024) framework.

As can be seen, Taipan consistently outperforms the baseline models across most tasks for all model sizes. Notably, the performance gap widens as the model size increases, with the 1.3B Taipan model showing significant improvements over other baselines. This suggests that Taipan's architecture effectively captures and utilizes linguistic patterns, even in tasks that do not fully showcase its long-context modeling capabilities.

## 4.3 IN-CONTEXT RECALL-INTENSIVE PERFORMANCE

To evaluate Taipan's proficiency in precise in-context retrieval, we assessed all models on a set of recall-intensive tasks Arora et al. (2024). These tasks are designed to test a model's ability to extract and utilize information from longer contexts, a capability particularly relevant to Taipan's architecture. Our evaluation suite includes two types of tasks: structured information extraction and question answering. For structured information extraction, we used the SWDE and FDA tasks Arora et al. (2024), which involve extracting structured data from HTML and PDF documents, respectively. To assess question-answering capabilities, we employed SQuAD Rajpurkar et al. (2018), which requires models to ground their answers in provided documents.

| Params | Model | SWDE | FDA | SQuAD | Avg. |
|---|---|---|---|---|---|
| 450M | Transformer++ | **43.0** | **48.7** | **18.1** | **36.6** |
| | Mamba | 27.9 | 9.8 | 12.5 | 16.7 |
| | Jamba | 35.4 | 36.6 | 16.3 | 29.4 |
| | **Taipan** | *41.4* | *39.6* | *17.8* | *32.9* |
| 1.3B | Transformer++ | **64.2** | **64.5** | **41.2** | **56.6** |
| | Mamba | 48.6 | 32.3 | 31.2 | 37.4 |
| | Jamba | 56.4 | 49.7 | 33.4 | 46.5 |
| | **Taipan** | *61.5* | *59.7* | *36.9* | *52.7* |

Figure 4: Performance on in-context retrieval tasks.

Table 4 demonstrates Taipan's significant performance advantages over both Mamba and Jamba in in-context retrieval tasks. Notably, Taipan achieves this superiority while consuming fewer computational resources than Jamba, which utilizes full attention mechanisms. This efficiency is attributed to Taipan's architecture, which combines Mamba-like elements with selective attention mechanisms, allowing it to filter out less important features. We also notice that Transformers excel at memory-intensive tasks in this experiment; however, they are constrained by linear memory scaling with sequence length, limiting their effectiveness and applicability for very long sequences. In contrast, Taipan maintains constant memory usage, offering a more efficient solution for processing long documents.

## 4.4 LONG-CONTEXT EXTRAPOLATION

Figure 1 illustrates Taipan's superior performance in handling extended sequences compared to Transformer, Jamba, and Mamba models. In perplexity evaluations across context lengths from 1K to 1M tokens (Figure 1a), Taipan yields the lowest perplexity, particularly excelling beyond the training context length. This performance contrasts sharply with other models: Transformers struggle with longer contexts due to quadratic computational complexity and linear memory scaling with sequence length, often leading to out-of-memory errors. Jamba, despite its hybrid nature, faces similar challenges due to its use of full attention mechanisms. Both Transformer and Jamba models exhibit limited extrapolation ability beyond their training context lengths. Mamba, while more efficient than Transformers and Jamba, still shows performance degradation for very long sequences.

Latency comparisons (Figure 1b) further highlight Taipan's exceptional efficiency. It demonstrates the lowest latency among all models, with linear scaling across sequence lengths. This contrasts with the quadratic scaling of Transformers and higher latency growth of Jamba. Notably, Taipan consistently outperforms Mamba-2, primarily due to its selective attention mechanism.

---

[1]https://github.com/EleutherAI/lm-evaluation-harness

## 5    ABLATION STUDY

We conducted a comprehensive ablation study to investigate the effect of the two key components in Taipan's architecture, i.e., the attention budget capacity $C$ and the inclusion of Positional Embeddings in the SALs, on its performance and efficacy.

### 5.1    EFFECT OF ATTENTION BUDGET CAPACITY

Our first experiment aimed to determine the optimal value of Capacity $C$ that would maintain computational efficiency while maximizing performance on downstream tasks. We trained multiple variants of Taipan, each with 1.3B parameters, using different Capacity $C$ values: 0.10, 0.15, 0.20, and 0.25. Each variant was trained for $24,000$ steps, allowing us to observe both the immediate impact of different $C$ values and their effect on model performance over time.

We evaluated the performance of each variant at regular intervals on two representative tasks: SWDE Arora et al. (2024) (for structured information extraction) and HellaSwag Zellers et al. (2019) (for commonsense reasoning). These tasks were chosen to assess both the model's ability to handle long-context retrieval and its general language understanding capabilities.

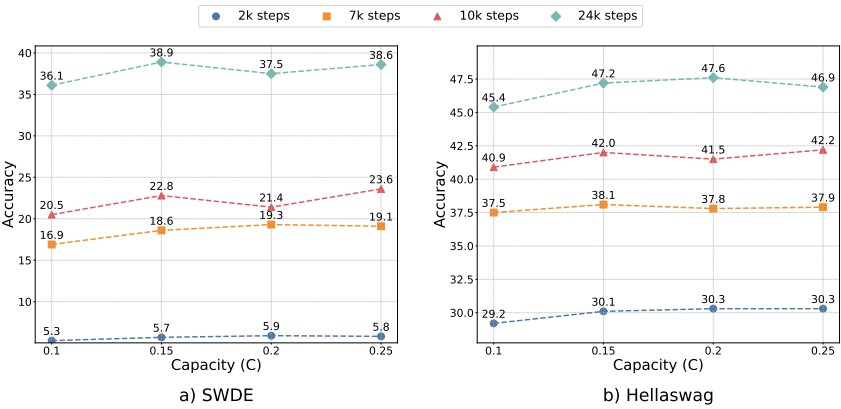

Figure 5: Effect of Attention Budget Capacity $C$ on Taipan's Performance

As illustrated in Figure 5, Taipan achieves optimal performance with a Capacity $C = 0.15$. We observed that increasing $C$ beyond 0.15 does not lead to significant improvements in results while increasing computational costs. Conversely, reducing $C$ below 0.15 resulted in a noticeable drop in performance on tasks requiring precise in-context retrieval or complex long-range dependencies. These findings support our hypothesis that computational demands vary across tokens, with many adequately represented by Mamba's Markovian structure without requiring attention mechanisms. By selectively applying attention only to tokens that benefit from it, Taipan optimizes resource allocation, enabling high performance while improving computational efficiency.

### 5.2    IMPACT OF POSITIONAL EMBEDDINGS

Our second experiment investigated the impact of Positional Embeddings in Taipan's Attention mechanism, focusing on the model's ability to handle and generalize to various context lengths. We trained two variants of the 1.3B parameter Taipan model for $24,000$ steps with a fixed context length of $4096$ tokens. One variant incorporates Rotary Positional Embeddings Su et al. (2024) in the Selective Attention layers, while the other excludes them. Figure 6 illustrates the performance of both variants in terms of perplexity across different context lengths.

The results reveal that Taipan without Positional Embeddings performs superiorly in generalizing context lengths beyond the training context. Both variants show comparable performance for sequences similar to or shorter than the training context length. However, as the sequence length increases, the performance gap between the two variants widens, with Taipan without Positional Embeddings maintaining lower perplexity scores. This suggests that the absence of Positional Em-

beddings enables more robust scaling to longer sequences. We attribute this improved generalization to the model's increased reliance on attention representation rather than positional biases.

# 6 RELATED WORK

Our approach builds on a foundation of relevant previous research. We will now discuss key studies that inform our methodology.

**State Space Models**: SSMs have emerged as a promising approach in attention-free architectures for language processing tasks. These models offer improved computational and memory efficiency compared to traditional attention-based models. The development of SSMs has progressed through several key iterations: S4 Gu et al. (2021a) introduced the first structured SSM, focusing on diagonal and diagonal plus low-rank (DPLR) structures. Subsequent variants like DSS Gupta et al. (2022), S4D Gu et al. (2022), and S5 Smith et al. (2023) improved on this foundation. Frameworks like GSS Mehta et al. (2023), H3 Fu et al. (2023), and RetNet Sun et al. (2023) incorporated SSMs into broader neural network architectures, often combining them with gat-

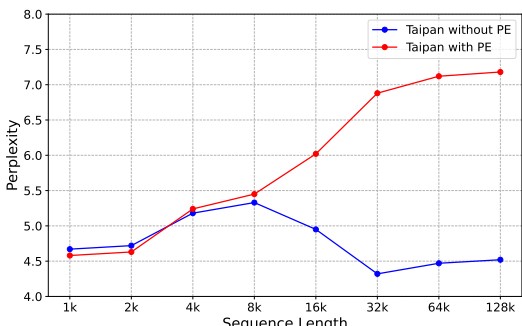

Figure 6: Perplexity comparison of Taipan variants with and without Positional Embeddings across different context lengths. Lower perplexity indicates better performance.

ing mechanisms or efficient attention approximations. Recently, Mamba Gu & Dao (2023) introduced time-varying or selective SSMs, which addresses limitations of static dynamics in previous SSMs by incorporating input-dependent state transitions, leading to improved performance in various tasks.

**Hybrid Architecture**: Several recent studies H3 Fu et al. (2023), Griffin De et al. (2024), Zamba Glorioso et al. (2024), Jamba Lieber et al. (2024) suggest the potential of blending SSM and the attention mechanism. These hybrid designs show promise in outperforming both traditional Transformers and pure SSM architectures, such as Mamba, particularly in scenarios requiring in-context learning capabilities.

**Long Context Models**: Recent advancements in sequence modeling have pushed the boundaries of context length, each with distinct approaches and challenges. Recurrent Memory Transformer Bulatov et al. (2023) demonstrated 1M token processing, but primarily on synthetic memorization tasks. LongNet Ding et al. (2023) proposed scalability to 1B tokens, yet practical evaluations were limited to sequences under 100K tokens. Hyena/HyenaDNA Poli et al. (2023); Nguyen et al. (2023) claimed 1M token context, but faced efficiency issues at longer lengths. Mamba Gu & Dao (2023) showed consistent improvements up to 1M tokens in DNA modeling and competitive performance across various language tasks.

# 7 CONCLUSION

Taipan presents a significant advancement in long-context language modeling by combining the efficiency of Mamba with strategically placed Selective Attention Layers. Our experiments demonstrate Taipan's superior performance across various scales and tasks, particularly in scenarios requiring extensive in-context retrieval, while maintaining computational efficiency. A key insight is that not all tokens require the same computational resources. Taipan's architecture leverages this observation through its selective attention mechanism, which dynamically allocates computational resources based on token importance. This hybrid approach addresses limitations of both Transformers and SSMs, offering a promising solution for efficient, large-scale language processing. Future work could explore further optimizations and applications of this architecture.

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

## A  DATASETS

Our training data comprises a diverse set of datasets, carefully curated to ensure breadth and depth across various domains. This diverse collection includes specialized mathematics datasets (Meta-MathQA Yu et al. (2023), NuminaMath-CoT LI et al. (2024), OpenWebMath Paster et al. (2023), Orca-Math Mitra et al. (2024)), high-quality web data (Fineweb-Edu-dedup Ben Allal et al. (2024)), synthetic data (Cosmopedia-v2 Ben Allal et al. (2024)), code data (Starcoderdata-python-edu Li et al. (2023)), and general knowledge sources (Wikipedia). This comprehensive approach aims to enable our model to handle a wide array of language modeling tasks. The inclusion of both domain-specific and broad-coverage datasets is designed to enhance the model's versatility and robustness across language modeling tasks.

All datasets were tokenized using the LLama3's tokenizer Dubey et al. (2024), resulting in 300B tokens.

The training data size varies by model scale: the 190M model is trained on 27 billion tokens (exclusively from Cosmopedia-v2), while the 450M and 1.3B models are trained on 100 billion tokens sampled from the combination of datasets mentioned above. Below are detailed descriptions of each dataset used:

1. **MetaMathQA** Yu et al. (2023): A comprehensive mathematics dataset designed to enhance the model's mathematical reasoning and problem-solving abilities.

2. **NuminaMath-CoT** LI et al. (2024): A chain-of-thought mathematics dataset that promotes step-by-step reasoning in mathematical problem-solving.

3. **Cosmopedia-v2** Ben Allal et al. (2024): A large-scale synthetic dataset for pre-training, consisting of over 39 million textbooks, blog posts, and stories.

4. **Fineweb-Edu-dedup** Ben Allal et al. (2024): A high-quality subset of the FineWeb-Edu dataset, containing 220 billion tokens of educational web pages. This dataset was filtered using an educational quality classifier to retain only the most valuable educational content.

5. **OpenWebMath** Paster et al. (2023): A diverse collection of mathematical content from over 130,000 different domains, including forums, educational pages, and blogs. It covers mathematics, physics, statistics, computer science, and related fields.

6. **Starcoderdata-Edu** Li et al. (2023): A subset of the Starcoder dataset, specifically filtered for high-quality educational content related to Python programming. This dataset aims to enhance the model's coding capabilities.

7. **Orca-Math** Mitra et al. (2024): A dataset focused on mathematical word problems, designed to improve the model's ability to interpret and solve practical mathematical scenarios.

8. **Wikipedia**: An English Wikipedia dataset providing a broad range of general knowledge across various topics.

