# OpenReview forum: "Taipan: Efficient and Expressive State Space Language Models with Selective Attention"
_ICLR.cc/2025/Conference — Submitted to ICLR 2025_

### Official Review · Reviewer_CMJ8 · 2024-10-30

**Soundness:** 3
**Presentation:** 3
**Contribution:** 2
**Rating:** 5
**Confidence:** 4

**Summary:**

The paper presents a hybrid architecture that combines Mamba-2 layers with softmax selective attention layers for long sequence modeling. The empirical study validates the effectiveness of the new model up to 1B model parameters.

**Strengths:**

The paper is well organized, and the research is well motivated.

**Weaknesses:**

The proposed hybrid model is conceptually similar to other hybrid models that combine softmax attention models (Transformers) and modern CNN layers (such as S4, Mamba). Although the gain on small models is encouraging, the paper could be much stronger if a more comprehensive comparison with SOTA Transoformer / Hybrid models can be performed.

**Questions:**

It is useful to run ablation experiments to show (1) the gain of using sliding window attention; (2) the gain over using only sliding window attention without selective attention; and (3) the number of tokens selected based on Eq. (1) and (2).

It is also useful to investigate what tokens are selected, and whether there are any patterns, such as the ones described in https://arxiv.org/pdf/2310.01801.

---

### Official Review · Reviewer_BrUv · 2024-11-03

**Soundness:** 1
**Presentation:** 2
**Contribution:** 2
**Rating:** 3
**Confidence:** 5

**Summary:**

The paper proposes to sparsify the query of self-attention layers in the context of layerwise hybridization between state-space models and self-attention.

**Strengths:**

1. The proposed approach is simple and shows more stable extrapolation on perplexity compared to Jamba.
2. The feature refinement mechanism, which uses the underlying probability distributino of the gumbel softmax to interporlate the residual branch output and the layer input, looks interesting.

**Weaknesses:**

1. The comparison between Jamba and Taipan is not fair: Taipan uses 1:6 for number of attention layers v.s. number of Mamba layers, while Jamba uses 1:7. Also, Taipan uses Mamba 2 while Jamba uses Mamba 1. The performance gain of Taipan in Table 1 can be from the fact that Taipan uses Mamba 2 and has more attention layers, and may have nothing to do with the proposed selective attention.
2. Lack of novelty: Hybridization between state-space-models and dynamic selective SWA has been explored in SeqBoat [1], but the paper does not include any discussions or emipical study to compare different selction mechanisms. Also, Taipan does not select key-value pairs, which will limit its long context performance.
3. Lack of important baselines: The paper should at least compare the performance of Taipan with a simple baseline that has 1:6 SWA-Mamba2 ratio to prove the effectivnees of the proposed selective attention. A more thorough comparisons should includes different sparse attention baselines as proposed in BigBird [2] and LongFormer, which are now well supported by FlexAttention [3].
4. Lack of implementation details: The paper does not includes a detailed description of how hyperparameters are configurated, such as: the temperature of Gumbel softmax, and how the query selection is efficiently implemented so that the proposed selective attention can result in wall-time speed up.
5. Taipan only shows non-exploading perplexity for long context extrapolation, which is trival for SWA based Mamba hybrid models, considering that Samba [4] already shows improving perplexity up to 1M context length. The paper can be strengthened with more evidiences on long context tasks such as Passkey Retrieval.


Missing References:

[1] Sparse Modular Activation for Efficient Sequence Modeling (NeurIPS 2023)

[2] Big Bird: Transformers for Longer Sequences (NeuIPS 2020)

[3] https://github.com/pytorch-labs/attention-gym

[4] Samba: Simple Hybrid State Space Models for Efficient Unlimited Context Language Modeling (arXiv 2023)

**Questions:**

1. Line 337: Jamba does not have positional embedding.

2. How is the model performance sensitive to the $\lambda$ of the constraint loss?

---

### Official Review · Reviewer_v4EA · 2024-11-04

**Soundness:** 3
**Presentation:** 3
**Contribution:** 2
**Rating:** 5
**Confidence:** 4

**Summary:**

This paper presents a hybrid approach that combines elements of Mamba and Transformer architectures, aiming to address two major challenges: the high computational complexity of Transformers in handling long contexts and the quality degradation issues encountered with Mamba. This approach aligns with prior research, including methods like Samba and Jamba.

The key contribution of the paper is its selective mechanism for token attention calculation. By incorporating a gating network, the model selectively skips attention computation for certain tokens, reducing inference costs. This optimization enhances the efficiency of the attention layer, achieving a notable speed-up without compromising performance and the paper demonstrates this with empirical results.

**Strengths:**

1. The paper’s motivation is clear, focusing on an important and timely topic with practical significance.
2. The writing is clear and well-organized, making it easy to understand.
3. The selective attention concept is well-founded and adds a valuable perspective to the field

**Weaknesses:**

The concept of selective attention is promising, as it aligns well with recent advances in efficient language models. However, similar approaches have been explored in prior work, including "Power-BERT: Accelerating BERT Inference via Progressive Word-vector Elimination" and "A Gated Self-attention Memory Network for Answer Selection." These studies also leverage selective focus on important tokens, prioritizing computation for tokens requiring additional context. Further distinction from these works, especially in terms of innovation and unique contributions, would enhance the impact of this paper.

Among previous research, "Samba: Simple Hybrid State Space Models for Efficient Unlimited Context Language Modeling" appears most comparable due to its hybrid structure and sliding window mechanism in attention. I would anticipate that Samba could achieve similar results in performance and latency to the model in this paper. A thorough empirical comparison with Samba would be beneficial to underscore the advantages and trade-offs of the proposed approach.

In Figure 1, perplexity seems to increase steadily from the start. Typically, one might expect an initial decrease in perplexity with context length before a rise as the length extends beyond a certain threshold such as the pre-training context length. Additionally, the claim that Taipan outperforms Mamba in terms of latency is unclear. Providing further clarification on latency measurements and factors contributing to Taipan’s efficiency would enhance the reader’s understanding.

Regarding task performance, additional explanation is needed to clarify why Taipan outperforms Transfer on the tasks listed in Table 1, as many involve short-context scenarios. More supporting evidence to validate Taipan’s superiority on these tasks would strengthen the claims. Furthermore, including benchmarks on MMLU and GSM-8K, which require higher reasoning capabilities, would offer a more comprehensive assessment of the model's generalization and reasoning skills.

**Questions:**

see weaknesses

---

### Official Review · Reviewer_BRJK · 2024-11-04

**Soundness:** 3
**Presentation:** 3
**Contribution:** 2
**Rating:** 5
**Confidence:** 4

**Summary:**

This paper proposes a new hybrid architecture that combines the recurrent formulation of state-space models with selective attention layers (SALs). The key component introduced, SAL, identifies tokens that have long-context dependencies, refines their features and augments their representations with standard attention. This aims to increase performance on tasks that require long-context memory without incurring the quadratic cost in standard Transformers. The evaluation performed on various tasks  and extrapolation shows superior performance compared to standard (Transformer++), efficient (Mamba) and hybrid (Jamba) models.  In addition, performance on recall-intensive tasks and model sizes up to 1.3B are also encouraging.

**Strengths:**

1. Existing hybrid architectures that combine state-space models with standard attention typically are applied to a small subset of layers for all tokens. The idea of applying attention only to a specific subset of positions dynamically through selective attention is novel and provides a more efficient way to augment recurrent models with attention.
2. The proposed model outperforms consistently two strong baselines that represent efficient and hybrid models, namely Mamba and Jamba, on general and recall-intensive tasks. At the same time, it outperforms standard attention models represented by Transformer++ or is behind by a moderate margin (~10% relative in recall-intensive tasks).
3. Apart from the quality, the proposed model has superior extrapolation capabilities up to 1M tokens and lower latency with increasing context size compared to the aforementioned baselines.

**Weaknesses:**

1. Even though the goal of selective attention is to improve efficiency, the experimental section does not quantify the computational benefits in terms of memory and latency compared to full attention or different budgets in practice. I'd suggest extending the experiment in Figure 5 to include memory use and training/inference times.
2. The comparison to previous efficient and hybrid models has limited coverage as it included only two baseline models and model sizes up to 1.3B. This reduces the potential impact of the main findings. To strengthen the claims regarding scaling, I'd suggest adding a larger model to reach the 7B mark and including a table with results compared to other recent efficient or hybrid architectures such as RecurrentGemma.
3. The experiment scope could benefit from recent general evaluation benchmarks for LLMs (MMLU, HELM, BBH), and instruction tuning or preference optimization experiments, with higher priority on the general evaluation.  The effect of different hyper-parameters such as sliding window size from 64 up to maximum context  length in a controlled experiment would also be useful.

**Questions:**

1. What is the computational benefit for different attention budgets and usage in different layers compared to full attention in terms of memory and latency? It would be useful if the authors provide some empirical evidence to quantify the benefits of SALs.
2. Could the authors include a few additional baselines in the datasets under study? I'd suggest to report scores in a table from prior work with efficient or hybrid architectures on the same datasets (e..g RecurrentGemma).
 3. The comparison to a baseline that uses only sliding window attention with the same window size as Tapain is missing. Could the authors report scores for this baseline across the tasks used in the experiment section? This would help to better understand the impact of selective attention.

---

### Official Review · Reviewer_U9JN · 2024-11-05

**Soundness:** 2
**Presentation:** 1
**Contribution:** 2
**Rating:** 3
**Confidence:** 4

**Summary:**

The paper presents Taipan, a hybrid model that incorporates attention modules into  Mamba-2 (an SSM). Specifically, it proposes to use Selective Attention Layers (SALs) to manage long-context tasks more efficiently in language modeling, such that only selected tokens are passed to (windowed) attention modules. In that way, Taipan selectively attends to critical tokens within an input, allowing it to capture long-range dependencies while also seeking to maintain computational efficiency.

**Strengths:**

I enjoyed the effort put in this paper towards balancing Mamba-2's efficiency with selective attention mechanisms, an approach that can offer benefits for handling long contexts. I also liked that the experimental setup contains multiple evaluations across various benchmarks and model scales, allowing some insight into Taipan’s potential in extended context scenarios.

**Weaknesses:**

I believe the paper has several notable weaknesses that limit its impact.

**Efficiency**: First, the presentation of efficiency gains is potentially misleading in Figure 1b, as Taipan’s backbone, Mamba-2, is slower than Taipan itself. Either that line represents Mamba-1, or the plot should include Mamba-2. To make matters more confusing, line 428 states, "Notably, Taipan consistently outperforms Mamba-2, primarily due to its selective attention mechanism." Therefore, how is it possible for a model that uses Mamba-2 to process the input, along with additional computations, to actually be faster than Mamba-2?  Overall, this discrepancy raises questions about whether computational overheads are fully accounted for.

**Novelty:** Furthermore, the novelty of combining SSMs with attention mechanisms is limited, as previous models, such as Jamba, have explored similar hybrid architectures, while the selective attention mechanism can be seen as an increment over Jamba.

**Gumbel-softmax**: Arbitrary architectural choices, like the selection of Gumbel-softmax without justification or comparison with alternatives, also weaken the paper, especially given that SALs are a primary contribution. The fixed attention capacity $C$ set during training could reduce the model’s flexibility at test time, as the need for attention across tokens may vary, and it is unclear how the model avoids processing all tokens at test time (as $C$ is budget for training).

**Presentation**: Additionally, inconsistencies in result reporting (e.g., bolded Taipan results even where it does not outperform other models) could mislead readers, as could unclear visual elements like Figure 1’s unexplained extrapolation regime and Figure 4’s table format. Moreover, the paper also disregards the proper use of citation styles (citep vs citet). Regarding Figure 1, it is unclear where the extrapolation regime starts, as per section 4.4. Collectively, these issues make the paper feel overly incremental and poorly substantiated.
Therefore, to improve the paper, I suggest consistently bolding the best results, clearly highlighting the extrapolation regime in Figure 1, improving Figure 4, and fixing the citation format throughout the paper.

**Questions:**

- Can you clarify which version of Mamba was used in Figure 1b? Can you provide a more detailed breakdown of the computational costs for both Taipan and Mamba-2?

- How does Taipan avoid the risk of all tokens being passed to the attention module at test time if the fixed attention capacity C is exceeded?

- Can you provide a rationale for choosing Gumbel-softmax? What would be potential alternatives? For example, how does Taipan compare with other differentiable strategies, such as gradient surrogates, continuous relaxations, etc, which have been shown to be effective in similar applications? See [1] for a comprehensive overview.

[1] Discrete Latent Structure in Neural Networks (https://arxiv.org/abs/2301.07473)

---

### Meta-Review · Area_Chair_Cfom · 2024-12-19

**Metareview:**

The paper addresses an important challenge in efficient long-context language modeling with a hybrid architecture that combines state-space models and selective attention. While the proposed approach has merit, limited baseline comparisons, and insufficient empirical evidence reduce the overall impact, e.g. the experiments lack critical ablations of core components and analysis of memory and compute usage. I recommend rejecting the paper in its current form.

**Additional Comments On Reviewer Discussion:**

The reviewers largely agreed on the relevance and ambition of the problem tackled by Taipan, a hybrid language model combining state-space models (SSMs) with selective attention layers (SALs) to efficiently handle long-context sequences. However, reviewers generally raised concerns about paper's novelty, soundness, and experimental designs.

To summarize, reviewers are concerned with (1) lacking solid baseline comparisons, e.g. BigBird, LongFormer (2) lack of quantification of compute and memory (3) lack of ablation experiments on core components.

These points are not addressed by the authors.

---

### Decision · Program_Chairs · 2025-01-22

Reject